# Exploiting of Secondary Raw Materials from Fish Processing Industry as a Source of Bioactive Peptide-Rich Protein Hydrolysates

**DOI:** 10.3390/md19090480

**Published:** 2021-08-25

**Authors:** Girija Gajanan Phadke, Nikheel Bhojraj Rathod, Fatih Ozogul, Krishnamoorthy Elavarasan, Muthusamy Karthikeyan, Kyung-Hoon Shin, Se-Kwon Kim

**Affiliations:** 1Network for Fish Quality Management & Sustainable Fishing (NETFISH), The Marine Products Export Development Authority (MPEDA), Navi Mumbai 410206, Maharashtra, India; girija_cof@yahoo.com; 2Department of Post Harvest Management of Meat, Poultry and Fish, Post Graduate Institute of Post-Harvest Management, Dr. Balasaheb Sawant Konkan Krishi Vidyapeeth, Roha 402109, Maharashtra, India; nikheelrathod310587@gmail.com; 3Department of Seafood Processing Technology, Faculty of Fisheries, Cukurova University, Adana 01330, Turkey; fozogul@cu.edu.tr; 4Fish Processing Division, ICAR-Central Institute of Fisheries Technology, Willingdon Island, Kochi 682029, Kerala, India; elafishes@gmail.com; 5The Marine Products Export Development Authority (MPEDA), Kochi 682036, Kerala, India; karthim05@gmail.com; 6Department of Marine Science & Convergence Engineering, Hanyang University, ERICA Campus, Ansan 11558, Gyeonggi-do, Korea; shinkh@hanyang.ac.kr

**Keywords:** fish proteins, bioactive peptides, fishery by-product, antioxidants, ACE inhibitory activity

## Abstract

Developing peptide-based drugs are very promising to address many of the lifestyle mediated diseases which are prevalent in a major portion of the global population. As an alternative to synthetic peptide-based drugs, derived peptides from natural sources have gained a greater attention in the last two decades. Aquatic organisms including plants, fish and shellfish are known as a rich reservoir of parent protein molecules which can offer novel sequences of amino acids in peptides, having unique bio-functional properties upon hydrolyzing with proteases from different sources. However, rather than exploiting fish and shellfish stocks which are already under pressure due to overexploitation, the processing discards, regarded as secondary raw material, could be a potential choice for peptide based therapeutic development strategies. In this connection, we have attempted to review the scientific reports in this area of research that deal with some of the well-established bioactive properties, such as antihypertensive, anti-oxidative, anti-coagulative, antibacterial and anticarcinogenic properties, with reference to the type of enzymes, substrate used, degree of particular bio-functionality, mechanism, and wherever possible, the active amino acid sequences in peptides. Many of the studies have been conducted on hydrolysate (crude mixture of peptides) enriched with low molecular bioactive peptides. In vitro and in vivo experiments on the potency of bioactive peptides to modulate the human physiological functions beneficially have demonstrated that these peptides can be used in the prevention and treatment of non-communicable lifestyle mediated diseases. The information synthesized under this review could serve as a point of reference to drive further research on and development of functionally active therapeutic natural peptides. Availability of such scientific information is expected to open up new zones of investigation for adding value to underutilized secondary raw materials, which in turn paves the way for sustainability in fish processing. However, there are significant challenges ahead in exploring the fish waste as a source of bioactive peptides, as it demands more studies on mechanisms and structure–function relationship understanding as well as clearance from regulatory and statutory bodies before reaching the end user in the form of supplement or therapeutics.

## 1. Introduction

Global fish production was estimated at 179 million tonnes in 2018, of which direct human consumption accounts for 156.4 million tonnes, with 22.2 million tonnes being utilized for non-food uses. Global per capita consumption of fish is 20.5 kg [1]. Fish is regarded as a high-quality protein source, which is consumed in either fresh or processed forms. More than 50–70% of total catch generated by fish processing is regarded as waste, also called secondary raw materials, which comprises diverse parts such as fish frames, visceral organs, skin, and heads [2]. Waste/ secondary raw material generated from processing is illustrated in Figure 1. Steadily increasing fish production leads to increased processing of fish and fishery products. 

Secondary raw materials are regarded as a rich source of bioactive components, with several applications in human nutrition and wellbeing [3]. Wider attention to the utilization of these materials could improve economics and sustainability at the same time [4]. There is a great possibility for the recovery of valuable biochemicals from secondary raw material and production of functional and biologically active foods.

### 1.1. Fishery By-Catch and Waste from Fish Processing

Currently, a large part of the total fish production is discarded or processed to fishmeal. Substantial amounts of fish and other aquatic species are processed, resulting in a large volume of by-products and wastes with higher levels of chemical oxygen demand (COD) and biological oxygen demand (BOD), which are likely to have negative impacts on coastal and marine ecosystems [5]. Significant amounts of fish processing waste generated from processing industries lead to higher disposal costs with greatly diminished economic returns. The use of fish by-products is an important production opportunity for the fishing and seafood processing industries since it can possibly create additional profits while also lowering disposal costs [6]. Benefits of utilizing secondary raw materials from fish processing are depicted in Figure 2. One of the potential possibilities for greater benefit is to use these industrial wastes and low-value fish for the recovery and hydrolysis of proteins rich in bioactive peptides [7]. Many biologically active peptides with valuable nutritional and functional qualities are found in hydrolysates, derived from the enzymatic hydrolysis of fish proteins [8]. Peptides with antioxidative, antihypertensive, and anticancer characteristics have been found in secondary raw materials originating in fish [9,10]. Anticoagulant and antiplatelet effects are also demonstrated by bioactive peptides derived from fish muscle [11]. Secondary raw materials supplied from aquatic sources can be processed using chemical/enzymatic/fermentation process to recover biochemicals [12].

Proteins and natural peptides occurring in the food system and living organisms play a vital role in the functioning of various systems including the cardiovascular, nervous, gastrointestinal, and immune systems. Peptides with biological or nutritional qualities help an organism’s body function better and increase the quality of its diet. These peptides are known as “bioactive peptides” [13], and these are formed of short sequences of amino acids including peptides having two or more amino acids which are inactive within the parent protein’s sequence and can be released through proteolytic hydrolysis with commercially available enzymes or proteolytic microorganisms as well as fermentation [14]. Enzymatic hydrolysis is a useful method for recovering essential components such as bioactive peptides from fish by-products, as well as improving the functional and nutritional qualities of protein without compromising its nutritional value [15]. The Hydrolytic breakdown of high molecular weight proteins to low molecular weight proteins is the basis of protein hydrolysate production [16]. Using different enzymes, different fish as substrates, and variable proteolytic factors such as pH, temperature, enzyme to substrate ratio, and time, a large range of hydrolysates with different physical, chemical, and biological characteristics can be produced [9]. Fish protein hydrolysates, like other relevant protein hydrolysates, can be used in food systems. 

### 1.2. Hydrolysis Mechanism

Proteins that are chemically or enzymatically broken down into peptides of various lengths are known as hydrolysates [17]. Protein hydrolysis improves the nutritional, functional, and bioactive aspects of fish proteins. Chemical hydrolysis involves the use of an acid or alkali to break down proteins, whereas enzymatic hydrolysis involves the use of enzymes from diverse sources to break down peptide links in proteins. Proteases, for example, catalyse only one specific process and work by building a complex with the substrate they are transforming. Proteases are classified further based on whether they hydrolyse the protein from the terminals (C-terminal or N-terminal) or in the core/middle of the protein chain as exopeptidases or endoproteinases, respectively. Exopeptidases cleave/ hydrolyse the terminal peptide bonds within protein molecules, usually at specific residues, to produce relatively small peptides, whereas endoproteinases cleave/hydrolyse the peptide bonds within protein molecules, usually at specific residues, to produce relatively large peptides [18].

In order to produce the FPH with different and desired properties, it is important to know the mechanism of protein hydrolysis. Enzymatic hydrolysis of fish proteins is a highly complex process, and its reaction mechanism has been poorly understood. Some proteases preferentially catalyze the hydrolysis of bonds adjacent to a particular amino acid residue, while some are less specific. The catalysis by proteases occurs primarily as three consecutive reactions [9]: (1) the formation of a complex between the original peptide chain and the enzyme referred to as the *Michaelis* complex; (2) cleavage of the peptide bond to liberate one of the two peptides; and (3) nucleophilic attack on the remains of the complex to split off the other peptide and to reconstitute the free enzyme. The generally accepted mechanism for proteases indicates that the dissociation of enzyme substrate complex into free enzyme and product is the rate-determining step, which in turn determines the overall rate of reaction. The hydrolysis of peptide bonds leads to an increase in the numbers of ionizable groups (NH3^+^ and COO^−^), with a concomitant increase in hydrophobicity and net charge, decrease in molecular size of the polypeptide chain, and an alteration of the molecular structure leading to the exposure of the buried hydrophobic residues to the aqueous environment [19,20,21].

When the degree of hydrolysis or the products formed are plotted against duration of hydrolysis, the hydrolysis is characterized by an initial rapid hydrolysis phase followed by a rapid decrease in the reaction rate. From the studies, the observed phase changes in hydrolysis curve have been ascribed to the phenomena like decrease in the concentration of peptide bonds, enzyme inhibition by hydrolysis products, and inactivation of the enzyme. The inhibition of the reaction by hydrolysis products found to have a significant effect on the hydrolysis curve. This is responsible for the typical shape observed during the enzymatic hydrolysis of fish by-product proteins. Removing the products formed during the hydrolysis process may improve the hydrolysis efficiency by the proteases. 

### 1.3. Influencing Factors in Hydrolysis

One of the most efficient approaches to recover powerful bioactive peptides is to use enzymes to hydrolyse fish waste proteins [22]. The ability of peptides to demonstrate bioactive qualities is determined by a number of factors. The protein supply, the specificity of the enzyme utilised in protein hydrolysis, the amino acid content, and the amino acid sequence in peptides sequence are important [14]. The reaction media’s physicochemical conditions including time, temperature, pH, and enzyme/substrate ratio have to be optimized by the enzyme’s activity. Proteases from a variety of sources are frequently utilised to achieve more selective hydrolysis since they are specific for peptide bonds next to specific amino acid residues. For the production of bioactive peptides from proteins, a variety of proteases has been utilised. Enzymatic protein hydrolysis has been achieved using proteases from animal, plant, and microbial sources. There are many scientific reports where different proteases from various sources have been used to produce bioactive peptides with desired health beneficial properties [23]. For the hydrolysis of tuna dark muscle [24], chymotrypsin, papain, neutrase, and trypsin at specific pH and temperature for each enzyme are used. The composition and sequence of amino acid residues in the peptides are considered to be directly linked to the biological activities [25]. The degree of hydrolysis (DH), as well as the size or molecular weight of the peptides, has been suggested as a contributing factor to the manifestation of bioactive properties [25]. The degree of hydrolysis is often used to evaluate the progress of hydrolysis reaction. However, for the given enzyme, substrate, and hydrolysis conditions, the composition, sequence, size/molecular weight of derived peptides, and, in turn, the bio-functionalities of protein hydrolysates derived from fish processing waste are all dictated by DH.

## 2. Fish Waste Protein Hydrolysates (FWPH)

Fish waste protein hydrolysate (FWPH) is a mixture of amino acids and peptides of varying sizes produced from secondary raw materials from fish processing, often by biological processes, that cleaves peptide bonds. Hydrolysis of proteins converts the secondary raw materials into a high value product which is rich in amino acids and biologically active peptides [26]. FWPHs have attracted significant attention due to their antioxidative, ACE-inhibitory, antibacterial, anticancer, and immunomodulatory activities, along with offering balanced essential amino acids as a source of bioactive components in food/feed. Proteases derived from plants, bacteria, fungi, and the viscera of fish can be exploited for making FWPH have good functionality and bioactive properties. Preparation steps involved in production of fish waste protein hydrolysate are given in Figure 3.

Protein hydrolysates and bioactive peptides were derived from various organs of aquatic organisms (Table 1) and have shown bioactive properties. Several authors have reported in vitro and in vivo bioactivities of peptides/protein hydrolysate derived from fish processing waste. Protein hydrolysates have been derived from head waste from different fish species including sardine, herrings, and salmon. Protein hydrolysates from other secondary raw materials like dark muscle from tuna and frame waste from Japanese threadfin breams, tilapia and other species have been studied for their bio-functional potential [27]. There are numerous reports available on hydrolysed proteins and bioactive peptides from skin (gelatin and collagen peptides) from a variety of fish species including catfish, croaker, horse mackerel, tilapia, cobia, carps and many more.

## 3. Bioactive Peptides

Bioactive peptides play a significant role in the digestive, cardiovascular, immune, neurological, and endocrine system. They are a novel source of biologically active regulators that can help prevent oxidation and microbial deterioration in foods, as well as aid the treatment of a variety of diseases and disorders, thereby improving the quality of human life [50]. Figure 4 shows bioactive and functional properties of peptides obtained from secondary raw materials from the fish processing industry.

Filleting waste produces a large number of fish frames, which are often discarded as industrial by-products by fish processing companies. Several researchers have attempted to produce protein hydrolysates from various by-products, including Alaska Pollack frame protein hydrolysed with mackerel intestine crude enzyme [29], yellowfin sole (*Limanda aspera*) frame protein hydrolysed with alcalase, neutrase, pepsin, papain and chymotrypsin

## 4. Bioactivities of Fish Waste Protein Hydrolysates

In this section, we have attempted to review some of the studies on bio-functionalities, including antihypertensive, antioxidant, antimicrobial, anticoagulant, and anticancer functionalities etc. that are reported for the protein hydrolysates/peptides derived from fish processing waste. 

### 4.1. Antihypertensive Properties of Fish Waste Protein Hydrolysates

Hypertension affects nearly a quarter of the world’s population and is a major, yet preventable, risk factor for cardiovascular disease and its consequences, which are the leading cause of death. Hypertension has a number of negative impacts on the human body, including an increased chance of coronary artery disease, stroke, cardiac arrhythmia, heart failure, and aberrant renal function as well as a slew of other issues associated with structural damage of the cardiovascular system [51]. Hypertension is mostly managed through a combination of lifestyle changes and antihypertensive medication [52,53]. The renin-angiotensin system (RAS), the kinin-nitric oxide system (KNOS), the rennin-chymase system (RCS), and the neutral endopeptidase system are the key physiological pathways involved in the control of blood pressure (BP) within the body (NEPS). The RAS is one of the most important humoral vasoconstrictor and vasodilator systems involved in blood pressure control. It controls blood pressure and electrolyte balance, as well as renal, neurological, and endocrine activities in the body that are related to cardiovascular regulation. 

#### The Renin-Angiotensin System’s Mechanism of Action and ACE-I Inhibition

Renin is released from the precursor molecule prorenin in the RAS by the action of kallikrein [54]. Angiotensinogen is cleaved by renin, releasing angiotensin I (Ang-I), a decapeptide (Asp-Arg-Val-Tyr-Ile-His-Pro-Phe-His-Leu). By eliminating the C-terminal dipeptide His-Leu, ACE converts angiotensin I into a potent vasoconstrictor, angiotensin II. In addition, ACE inhibits the vasodilator bradykinin [13]. As a result, inhibiting ACE is thought to be a useful strategy for treating hypertension. Although synthetic ACE inhibitors have shown to be effective, they come with a list of side effects that includes cough, loss of taste, renal impairment, and angioneurotic edoema [55]. As a result, for the prevention and treatment of hypertension, it is vital to look for natural, safe, and more effective ACE-inhibitory medicines as alternatives.

ACE inhibition is regarded as important in regulating blood pressure and treating hypertension [56]. Extensive uses of chemical-based medications have led to negative impacts on individual health. In order to avoid the adverse effects of synthetic drugs, researchers are looking for natural ACE inhibitors as alternatives [56]. ACE inhibitory (ACE-I) peptides from natural sources such as milk, eggs, meat, fish, and plants are now the focus of current research [57]. Bioactive peptides from fish and fishery products are showing higher ACE inhibition than many available natural ones [58]. Hydrolysates derived from fish waste have acted as biocatalysts converting angiotensin-I into -II, which is responsible for vasoconstriction [59]. ACE inhibitory peptides have been identified and described from a variety of fish sources, including muscle proteins and fish processing waste and discards. The ACE inhibitory action of several enzymatic hydrolysates derived from fish waste has been widely reported [42,60,61,62]. 

Peptides derived from fish waste hydrolysates have shown excellent ACE inhibition activity due to their high affinity for ACE active sites [63], blocking or altering the conformation of a site [64,65]. The higher affinity towards ACE sites and slower elimination of marine derived peptides, have made them more potent than commercial options [66]. Peptides from hydrolysed fish with molecular weight (>1500 Da), short chain length, C terminal peptide sequence, presence of lysine or arginine at C end, and hydrophobic amino acids showed higher ACE inhibitory activity regulating blood pressure [67,68,69,70,71]. Alaska pollock frame protein hydrolysates showed ACE inhibitory peptides for 1 kDa protein fraction [29], while dipeptide derived from sardine showed higher antihypertensive activity [72]. Peptides exhibiting ACE-I inhibitory activity are dipeptides such as glycine, lysine, arginine, threonine, valine with proline, threonine or valine with proline, glycine-isoleucine. and phenylalanine-asparagine. Table 2 summarises research on the antihypertensive characteristics of fish waste protein hydrolysates conducted by a few researchers. Antihypertensive properties are usually attributed to the amino acid sequence of peptides, hydrophobic properties, and chain length [73]. An ACE inhibitory peptide isolated from salmon skin showed a molecular weight of 770 Da with sequence Gly-Leu-Pro-Leu-Asn-Leu-Pro. It showed reduced systolic blood pressure in rats with oral administration [74].

### 4.2. Antioxidative Properties of Fish Waste Protein Hydrolysates

#### 4.2.1. The Mechanism of Oxidation and Its Impact on Food

The loss of electrons from a molecule, the addition of oxygen, and the loss of hydrogen from a compound [83] are necessary processes in aerobic organisms, especially in vertebrates and humans, despite the generation of free radicals. Unsaturated fats are oxidised largely through a free radical-mediated mechanism. Lipid peroxy radicals are formed when the radicals react with molecular oxygen. These radicals can take a hydrogen atom from nearby unsaturated fatty acids to form a hydroperoxide and a new lipid radical, causing the chain reaction to continue and accelerate. In the body’s regular usage of oxygen during respiration, the creation of reactive oxygen species (ROS) such as superoxide anion radicals (O_2_), hydroxyl radicals (OH), and non-free radical species such as hydrogen peroxide (H_2_O_2_) and singlet oxygen (^1^O_2_) is an unavoidable consequence [84]. Due to its potential negative impacts, such as off flavours, odours, dark colours, and the creation of potentially hazardous compounds, this process is particularly important in the food and health industries [85]. Researchers are interested in lipid oxidation, and they are working on how to avoid oxidation utilising bioactive peptides. Lipid oxidation degrades food quality by causing rancidity and the production of harmful chemicals in food systems. Lipid oxidation in food can be slowed by lowering metal ions and limiting exposure to light and oxygen through packaging, as well as utilising the right amount of natural and synthetic antioxidants.

#### 4.2.2. Antioxidants

Antioxidants are employed to reduce damage induced by free radicals, extend the shelf life of lipid-containing foods, and retain their nutritional quality, as well as to mitigate the effects of oxidative damage on the human body [86]. Synthetic antioxidants include butylated hydroxy anisole (BHA), tertiary-butyl-hydroquinone (TBHQ), and butylated hydroxytoluene (BHT), while naturally occurring antioxidants include tocopherol, ascorbate, and carotenoids. However, because of concerns about carcinogenic consequences, natural antioxidants from plant and animal sources, such as protein hydrolysates, peptides, and amino acids, are now being explored. In general, a peptide’s capacity to stabilise free radicals is related to its ability to donate or take electrons from the free radical in order to lessen its reactivity. Antioxidants are commonly included in dietary supplements to improve health and reduce the risk of diseases like cancer and heart disease.

#### 4.2.3. Antioxidative Properties of Fish Waste Protein Hydrolysates

Proteins from a variety of plant and animal sources, as well as their hydrolysis products, individual peptides, and amino acids, have been shown to have antioxidant action against lipid and fatty acid peroxidation. Based on the nature, size, and composition of the various peptide fractions as well as the protease specificity, peptides produced from enzymatic digestion of various fish proteins have been found to have antioxidant potential [87]. Lipid peroxidation in meals reduces nutritional value and increases the risk of sickness after consuming food that could induce a hazardous reaction [14]. The degree of hydrolysis and the enzyme type utilised influenced the antioxidative characteristics of round scad and yellow stripe trevally [88]. Amino acid composition and their sequence (hydrophobic and aromatic), hydrophobicity, chain length of peptides, and size (0.5–3 kDa) formed by hydrolysis have shown advantages over proteins considering the antioxidative activity [89,90]. High antioxidative peptide isolation from visceral organs of horse mackerel using gastrointestinal digestion showed better antioxidative action when compared with other natural antioxidants like tocopherol [91]. Table 3 summarises the antioxidant properties of fish protein hydrolysates. The bioactive characteristics of peptides were investigated, as was the sequencing of the peptides. 

Peptides derived from hydrolysed proteins are known to scavenge free radicals (proton donation), maintain redox balance and metal chelating activity (pro-oxidant), and inhibit enzymatic and non-enzymatic activity [71,88,101,102,103]. Alternately, it is reported that there are enhanced antioxidative activities of enzymatic and non-enzymatic enzymes derived from sardinella by-products protein hydrolysates [104]. The atomic hydrogen donating ability of amino acids such as histidine, leucine, tyrosine, methonine and cystenine exert antioxidative capacity [105]. The antioxidant activity is found to be highly variable based on enzyme and conditions of hydrolysis (degree of hydrolysis, time, temperature and pH) to produce hydrolysates [86,89]. Mendis and team [101] suggested hydrophobicity of the amino acids is also associated in providing antioxidative ability in peptides derived from jumbo squid (*Dosidicus gigas*) skin gelatin.

### 4.3. Anticoagulant Activities of Fish Waste Protein Hydrolysates

During blood coagulation, a sequence of processes occurs that, if not controlled, can lead to coronary artery occlusion. Numerous platelets assemble during platelet activation, which is followed by the conversion of prothrombin to thrombin, a serine protease that converts soluble fibrinogen to its insoluble form, fibrin. The conversion of fibrinogen to fibrin is facilitated as more thrombin is generated. Myocardial ischemia and heart attacks are caused by the combined effects of vasoconstriction and blockage of coronary arteries by fibrin complex development within the blood vessel. As a result, limiting platelet aggregation in these settings will diminish vasoconstriction and hence the risk of myocardial ischemia. Rajapakse and team [101] isolated anticoagulants from yellowfin sole protein hydrolysate. Regardless of Zn2^+^, it blocked activated coagulation factor XII by forming an inactive compound. Higher anticoagulation activity was found for peptides containing His-Cys-Phe, Cys-Leu-Arg, Leu-Cys-Agr-Agr, and Leu-Cys-Arg amino acids [93]. Adductor muscle from marine bivalve mollusc (*Mytilus edulis*) protein hydrolysates showed higher anticoagulation ability as 1.49 mg/mL [106]. Similarly, anticoagulant peptide from oyster (*Crassostrea gigas*) hydrolysate inhibited blood coagulation factor [107]. Oyster (*Crassostrea gigas*) hydrolysate and *Scapharca broughtonii* protein (26 kDa) were found to prolong the inhibition of thrombin time and activate partial thromboplastin time [107,108]. 

### 4.4. Antimicrobial Properties of Fish Waste Protein Hydrolysates

Antimicrobials in food are used to prevent the growth of microorganisms that cause food spoilage. The most frequently used antimicrobial drugs are organic acids (sorbic acid, acetic acid, citric acid etc.), modifying cell membrane permeability to substrates and establishing inhospitable pH conditions for bacteria growth [73]. When exposed to water, organic acids such as sorbic acid, despite being a highly good preservative, are known to degrade and release potentially hazardous chemicals such as acetaldehyde. As a result, using antimicrobial peptides is a possibly safer approach for preventing similar incidents. Peptides with varied amino acid sequences and varying sizes have been found to have antibacterial action after enzymatic hydrolysis of proteins. Anti-microbial peptides have been found to have a wide spectrum of activity against bacteria, viruses, fungus, and protozoa, among other microbes. They have a broad spectrum of antibacterial and antifungal activities. Antimicrobial peptides that have antimicrobial properties are a class of bioactive peptides and are produced naturally or from protein hydrolysis. 

Hydrolysed secondary raw materials from Atlantic mackerel (*Scomber scombrus*) [109] and half-fin anchovy (*Setipinna taty*) [110] showed antimicrobial activity against gram negative *Escherichia coli* and gram-positive *Listeria* innocua. Four (P4, P8.1, P8.2 & P11) antibacterial peptides were found in Atlantic mackerel by-products [108]. Antimicrobial peptide P4 showed complete inhibition of *Listeria innocua* and *Escherichia coli,* implicating the action of the amino acid sequence. The antimicrobial action (complete inhibition) of collagencin peptide extracted from fish collage against Staphylococcus aureus was reported by [111]. Partial action against *Staphylococcus pyogenes*, *Escherichia coli*, *Listeria innocua*, *Lactococcus lactis* and *Carnobacterium divergens* was reported. Peptide derived from by-product of snow crab hydrolysate showed antimicrobial activity, which was due to the low molecular weight of peptide (800 Da) [112]. 

The majority of antimicrobial peptides have been shown to be cationic, which means they have a net positive charge because of positively charged amino acid groups like lysine and arginine with hydrophobic properties and amphipathicity [110,113,114], which facilitate the bonding with the host and solubility in aqueous (lipid) membranes. Antimicrobial peptides are thought to work by forming membrane pores and then penetrating the cell, allowing microbial biological components to be released and the cell to be destroyed [110]. Antimicrobial activity is found in peptides which are released by appropriate hydrolysis conditions [110,115], usually with less than 50 amino acids and low molecular weight of <10 kDa [86].

### 4.5. Anticancer Activity of Fish Waste Protein Hydrolysates

Cancer has had a direct and indirect impact on the global population as a leading cause of death. While the number of cancer cases is increasing, some of them could be prevented or even treated with natural chemicals. It has been claimed that bioactive peptides found on land and in the sea can minimise the risk of chronic diseases and maintain good health. Since fish by-products as source of bioactive peptides have anticarcinogenic potential [116]. Anticarcinogenic peptides fight cancer cells in a variety of ways, including (1) in the cytoplasm, (2) through stimulation of membrane disruption by micellization, and (3) in the interaction of peptides with cells during apoptosis gangliosides on the surface [82]. Peptides derived from secondary raw material from fish processing demonstrated anticancer activities [67,117]. However, there is paucity of literature available on in vivo studies and cell line studies to evaluate the anticarcinogenic effect of peptides derived from waste of seafood processing. Table 4 shows the anticarcinogenic activity of peptides derived from secondary raw materials of aquatic origin.

Bioactive peptide from sepia ink hydrolysate, tuna dark muscle, flat head (*Platycephalus fuscus*), milk fish (*Chanos chanos*) skin collagen, and skate (*Raja porosa*) cartilage hydrolysate showed antiproliferative activity against prostate cancer (DU-145), breast cancer (MCF-7) cells, 29 colon cancer cells, and HeLa cells [24,118,121,127,128]. The antiproliferative effect of fish waste protein hydrolysates correlates with their antioxidant activities as demonstrated by [129] in oyster hydrolysates that had higher radical scavenging activity as well as anticancer activity against human colon cancer cell line. Presence of amino acids like Pro, Gly, Lys, Arg, and Tyr were known to be responsible for anticancer activity (apoptosis, antiproliferation, stimulation etc.) against colon, prostate, breast, gastric, and cervical cancer cell lines as reviewed in another study [130]. The lower molecular weight of free amino acids and peptides (300–1950 Da) imparted them with anti-cancerous activity, due to high mobility and diffusion [130,131]. Peptide derived from protein hydrolysates made up of hoki frames using enzymatic hydrolysis by different proteases like papain showed decreased cytotoxicity on human embryonic fibroblastic cells and protected from DNA damage [132].

### 4.6. Other Bioactive Properties of Fish Waste Protein Hydrolysates

#### 4.6.1. Anti-Inflammatory Property of Fish Waste Protein Hydrolysates

In infections and injuries, inflammation is an important mechanism promoting healing which also participates in immune mechanism, homeostasis, and cell functioning [133]. Due to the increased interest for anti-inflammatory compounds from natural sources with lesser side effects, many researchers are actively working in this area [134]. Hydrolysed proteins from fish waste and peptides derived also show property to fight against inflammation because of their lower molecular weights and short chain amino acid sequences with positive charge and presence of hydrophobic amino acids leading to radical scavenging mechanisms. In general, peptides having positively charged amino acids like lysine, arginine and histidine in C or N-terminal show anti-inflammatory properties [135]. Visceral protein hydrolysates from anchovy produced by protamex, flavorzyme, and alcalase exhibited stimulated anti-inflammatory properties in mouse macrophage cells for peptides identified as SNKGGGRPN, PGVATAPTH, and LLGLGLPPA [136]. Protein hydrolysates from bones of salmon and pectoral fins of salmon using papain and pepsin enzymes also showed anti-inflammatory effects [137,138]. 

#### 4.6.2. Calcium Binding Property of Fish Waste Protein Hydrolysates

Calcium forms an important part of the body of human beings and is generally obtained from dairy products such as milk [139]. Phosphopeptides of casein improve calcification of bones by binding with calcium and preventing Ca salt precipitation [140]. In terms of enhancing the bioavailability of calcium, aquatic sources provide one of the best alternatives to milk. Peptides of oligo nature extracted from bones of hoki fish showed calcium binding properties [141] that inhibited Ca salt formation. Polypeptide extracted from Alaska pollock bones Val–Leu–Ser–Gly–Gly–Thr–Thr–Met–Ala–Met–Ala–Met–Tyr–Thr–Leu–Val showed their affinity to Ca ions [142,143]. Peptide sequence of Hoki frame peptide was Val–Leu–Ser–Gly–Gly–Thr–Thr–Met–Tyr–Ala–Ser–Leu–Tyr–Ala–Glu, which exhibited a Calcium binding property [30]. Phosphopeptides of oligo nature have the potential to be used as nutraceuticals. 

#### 4.6.3. Wound Healing Property of Fish Waste Protein Hydrolysates

Proliferation and cell migration with new extracellular matrix formation are a few of the steps involved in the wound healing process [144]. Oral administration of peptides extracted from various fish species and their by-products, such as collagen hydrolysates, demonstrates retention of moisture over the face along with enhanced viscoelastic properties and reduced sebum levels [145]. Enzymatic protein hydrolysates derived from the bones of silver carp and isolated peptides showed higher efficiency in stimulating metabolism of keratinocytes and wound healing activities, demonstrating the promising nature of bone peptides in care of wounds in the cutaneous region [146].

#### 4.6.4. Neuroprotective Property of Fish Waste Protein Hydrolysates

Neuroprotection is one of the important identified functions of fish waste protein hydrolysates. The ability of bioactive peptides from enzymatically hydrolysed fish processing waste to act against neurodegradative diseases like Alzheimer’s disease is of great importance due to increased life expectancy in developed countries [147]. Few of the protein hydrolysates studied for neuroprotective activity are skate skin hydrolysates from which β-secretase inhibitory peptide was purified with an IC50 value of 24.26 μM [148]. Grass carp skin protein hydrolysates showed neuroprotective effects on induced cytotoxicity studies [149]. Mice studies for enzymatic salmon skin hydrolysates exhibited enhanced memory and learning [150].

## 5. Bioactivity of Fish Gelatin Hydrolysates

Fish skin gelatin is usually obtained using denaturation of proteins by heating, which has a wide range of applications in pharmaceutical, photographic, food and cosmetic applications [151]. In addition, gelatin can be hydrolysed to derive bioactive peptides using a hydrolysis mechanism. There are several reports available on fish skin gelatin hydrolysates production and characterization for their functional and bioactive nature [58]. The bioactive properties of fish gelatin protein hydrolysates from the skin of several species such as squid, Alaska pollock, red snapper, cobia, hoki, sole, etc. have been reported [152,153,154,155,156,157,158]. Peptides derived from hydrolysed fish skin gelatin acted as good antioxidative and antihypertensive compounds in vitro and in vivo. Serially hydrolysed Alaskan pollock gelatin hydrolysate showed antioxidative activity, and the peptide sequence identified consisted of glycine residues at the C-terminal end and sequences having Gly-Pro-Hyp [155]. Gelatin hydrolysates of fish skin exhibited better bioactive properties compared to fish muscle derived peptides [68,155]. A unique peptide sequence containing Gly-Pro-Ala in protein hydrolysates of fish gelatin demonstrated antioxidant activity along with increased calcium bioavailability [152,159]. Milkfish collagen hydrolysates prepared using bacterial proteases demonstrated antioxidant and antifungal activity against *Candida* sp. by 5 log cycle reduction in fungal load [160]. Hydrolysed scale gelatin from skipjack tuna showed a better antioxidant property for peptides having Gly in the sequence [161]. Intrinsic active properties of hydrolysed fish gelatin need more exploration as anticarcinogenic and antidiabetic agents [162]. Hydrolysed skin gelatins from skate exhibited antihypertensive effect for enzymatic hydrolysis with alcalase. Isolated and identified peptides had a sequence namely MVGSAPGVL (829 Da) and LGPLGHQ (720 Da), with IC_50_ values of 3.09 and 4.22 µM [163].

## 6. Research Gaps, Opportunities, and Challenges for Fish Protein Hydrolysate

The current interest in fish hydrolysates containing large amounts of protein and thereby peptides and amino acids is of great importance for future research. Although the functional and antioxidant characteristics of fish processing waste protein hydrolysates have been thoroughly reported [2,42], more study is needed in order to commercialise such hydrolysates while also taking into account the economic factors. To maintain the appropriate balance between the functional and bioactive qualities of these hydrolysates, the extent to which fish waste proteins can be hydrolysed must be optimised. There has not been enough reporting on the usage of protein hydrolysates in food ingredients. It is necessary to conduct additional research into the safety of these hydrolysates. Microbiological standards for fish protein hydrolysate have not been prescribed. However, there are limited studies on safety aspects of hydrolysates produced from fish processing waste and further research of a similar nature is required. Due to the inherent nature of process flow employed in hydrolysate production, bacteria that are more hazardous are killed during the enzymatic hydrolysis and post-hydrolysis processes; however, there is a scarcity of scientific evidence on the safety of fish waste protein hydrolysates. Before commercialization, microbial and allergen analyses are required to complete the food safety profile of protein hydrolysate. In addition, because these hydrolysates come from seafood, histamine levels must be monitored. Although many in vitro investigations on the bioactivity of protein hydrolysate have been undertaken, the destiny of these functional molecules in the gastrointestinal tract, as well as their absorption and bioavailability, has yet to be fully explored.

## 7. Conclusions

Large volumes of waste are generated during the processing of fish, which might be used to make beneficial fish protein hydrolysate. Researchers have focused their attention on bioactive protein hydrolysates from fish waste in order to develop natural alternatives to synthetic options. Temperature, hydrolysis time, enzyme to substrate ratio, pH, substrate used, and degree of hydrolysis are all elements that influence the hydrolysis process as well as the bioactive qualities of the fish waste protein hydrolysates produced. Under numerous brand names, fish protein hydrolysates are being utilised as health supplements or nutraceuticals. However, after a thorough assessment of the safety, the utilisation of fish waste for the synthesis of physiologically active protein hydrolysates is feasible on an industrial scale. Focus must also be placed upon quantification of unique bioactive peptides derived from fish protein hydrolysates in systematic manner.

## Figures and Tables

**Figure 1 marinedrugs-19-00480-f001:**
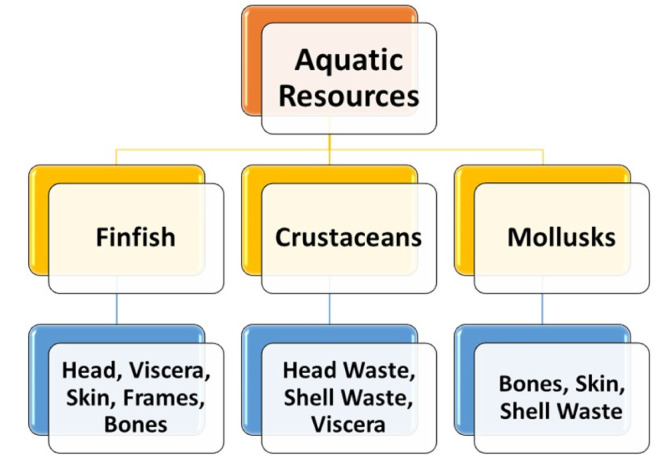
Categories of waste/secondary raw materials of aquatic origin.

**Figure 2 marinedrugs-19-00480-f002:**
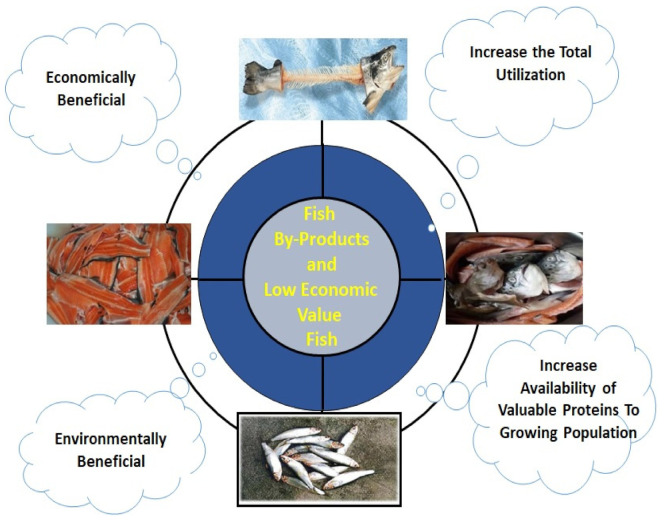
Benefits of harnessing secondary raw materials from fish processing.

**Figure 3 marinedrugs-19-00480-f003:**
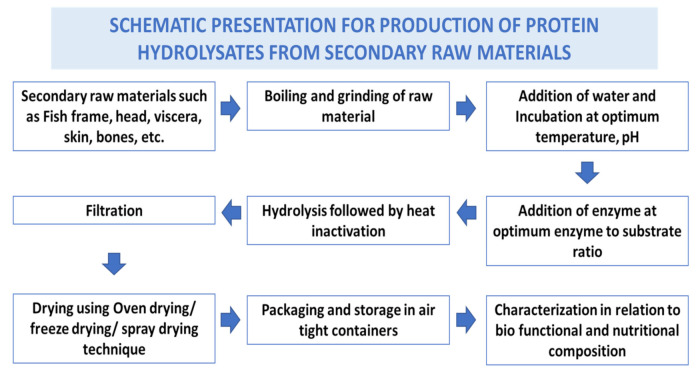
Schematic presentation for production of protein hydrolysates from secondary raw materials.

**Figure 4 marinedrugs-19-00480-f004:**
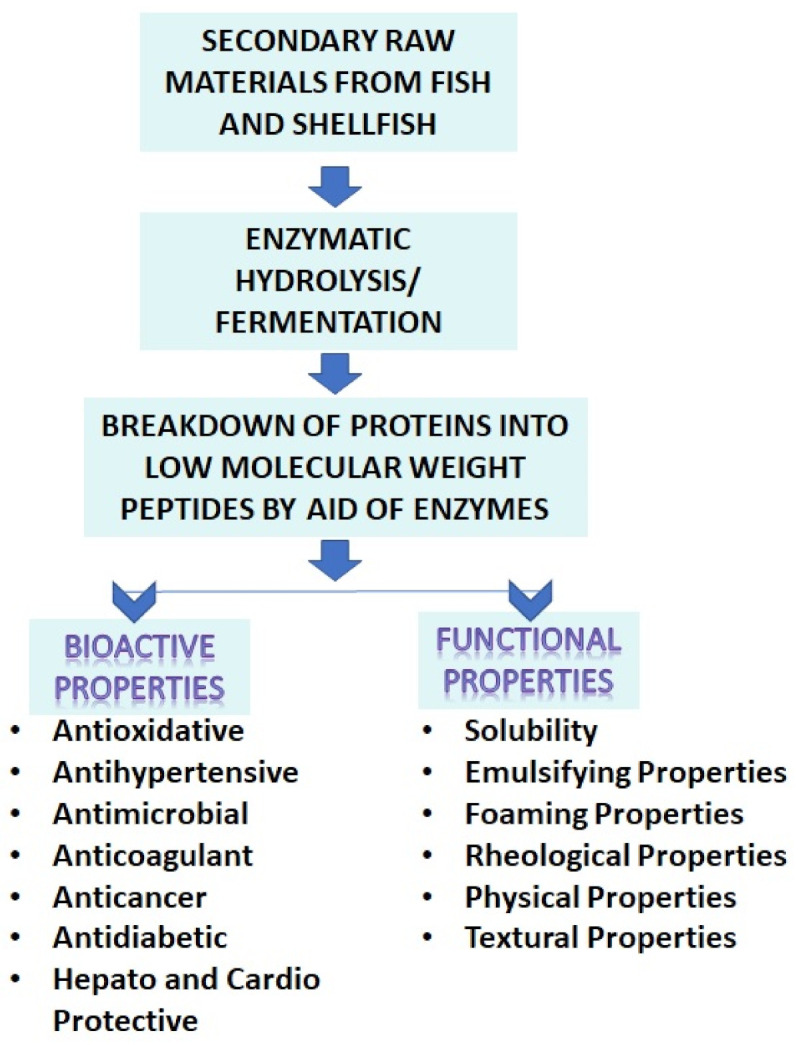
Bioactive and functional properties of peptides obtained from secondary raw materials from the fish processing industry.

**Table 1 marinedrugs-19-00480-t001:** Fish and secondary raw material used for deriving bioactive peptides.

Fish	Secondary Raw Material	Reference
Sea bream	Fish Scales	[28]
Alaska pollack (*Theragra chalcogramma*)	Frame	[29]
Hoki (*Johnius belengerii*)	Frame	[30]
*Catla catla*	Visceral organs	[31]
Sturgeon	Visceral organs	[32]
Tuna	Liver by-products	[33]
Alaska pollock	Frames/backbones	[34]
Skate	Skin	[35]
Cod (*Gadus macrocephalus*)	Skin	[36]
*Channa striatus*	Roe	[37]
Bluefin tuna	Head	[38]
Salmon	Pectoral fin	[39]
Seabass	Skin	[40]
Leatherjacket	Head waste	[41]
Japanese threadfin bream	Frame waste	[42]
Rainbow trout	By-products	[43]
Squid	By-products	[44]
*Leiognathus splendens*	By-catch	[45]
Salmon	By-products	[46]
Monkfish	By-products like head and viscera	[47]
Turbot	By-products	[48]
Fish	Solid and liquid waste generated from processing operations	[49]

**Table 2 marinedrugs-19-00480-t002:** Antihypertensive characteristics of fish waste protein hydrolysates.

Substrate	Enzymes	ACE Inhibitory Activity (%) or (IC_50_ Value)	Reference
Yellowfin sole frame	Chymotrypsin	Ultrafiltration fractionation- Fraction I: 47.6% Fraction II: 34.5% Fraction III: 68.8%	[61]
*Sardinella* byproducts	Protease-K	47.4% activity	[62]
	Alcalase	43.0% activity	
	Sardine visceral enzyme	63.2% activity	
	Chymotrypsin	55.8% activity	
	Protease ES1	13.2% activity	
Tilapia	Cryotin	62–71% activity	[75]
	Flavourzyme	66–73% activity	
Jelly fish	Papain	IC_50_: 6.56 µM	[76]
Freshwater clam byproducts	Pepsin Trypsin	IC_50_: 0.23 mg/mL	[77]
Giant Jelly fish	Alcalase	39.61% activity	[78]
	Flavourzyme	36.36% activity	
	Neutrase	62.29% activity	
	Papain	76.73% activity	
	Protamex	70.01% activity	
	Trypsin	68.01% activity	
Rohu roe	Pepsin	47% activity	[79]
	Trypsin	36% activity	
Pink perch fish frame waste	Papain Bromelain	69% activity	[42]
Salmon skin	Alcalse, papain	IC_50_: 60 µM	[70]
Sea bream scale	Protease	IC_50_: 7.5 µM	[28]
Skate (*Raja kenojei*)	Alcalase, α-chymotrypsin, neutrase, pepsin, papain and trypsin	IC_50_: 95 µM and 148 µM	[35]
Tuna (*Thunnus obesus*)	Alcalase, neutrase, pepsin, papain, α-chymotrypsin and trypsin	IC_50_: 11.28 µM	[63]
Tuna heads	Alcalase	0.27 mg/mL	[62]
Sardine viscera	Alcalase	1.16 mg/mL	[80]
Nile tilapia skin	Alcalase	1.12 mg/mL	[81]
Hound fish (*Caprosaper linnaeus*) viscera	Alkaline protease	75 µg/mL	[82]
Alaska Pollock skin gelatin extracts	Alcalase, pronase E, and collagenase	Gly-Pro-Leu and Gly-Pro-Met Peptides IC50: 2.6 and 17.13 µM	[68]

**Table 3 marinedrugs-19-00480-t003:** Antioxidant properties of fish waste protein hydrolysates.

Substrate	Enzymes	Bioactive Properties Studied and Peptide Sequence	Reference
Striped catfish frame meat	Papain Bromelain	DPPH radical scavenging activity (90%), ferric reducing antioxidant power	[92]
Goby muscle proteins	Alcalase	DPPH radical scavenging activity and reducing power	[93]
*Rastrelliger kanagurta* backbone	Pepsin Papain	DPPH radical scavenging activity (36–46%)	[94]
Salmon protein hydrolysate	Pepsin	DPPH radical scavenging activity (55%)	[95]
Pink perch frame waste hydrolysate	Papain Bromelain	DPPH free radical scavenging activity (up to 90%), ferric reducing antioxidant power	[42]
Salmon (*Salmo salar*) frame hydrolysates	Alcalase Papain	DPPH free radical scavenging ability, ABTS activity, ferric reducing antioxidant power (FRAP), metal chelating activity and oxygen radical antioxidant capacity (ORAC)	[96]
*Oreochromis niloticus* Scale gelatin hydrolysate	Alcalase Pronase E trypsin pepsin	DPPH radical scavenging activity, hydroxyl radical scavenging activity and superoxide radical anion scavenging activity	[97]
Black pomfret visceral protein hydrolysate	Pepsin trypsin ά-chymotrypsin	DPPH radical scavenging activity, FRAP and metal chelating activity (Ala-Met-Thr-Gly-Leu-Glu-Ala)	[98]
Alaska pollack frame protein hydrolysate	Mackerel intestine crude enzyme	Higher antioxidant Activity in terms of ferric thiocyanate for peptide fraction < 1 kDa Peptide sequence: Leu-Pro-His-Ser-Gly-Tyr (627 Da)	[29]
Tuna backbone protein hydrolysate	Alcalase, a-chymotrypsin, neutrase, papain, pepsin and trypsin	Higher lipid peroxidation inhibition and DPPH free radical scavenging activity for peptide—VKAGFAWTANQQLS (1519 Da)	[14]
Cod (*Gadus macrocephalus*)	Alcalase, neutrase, papain, trypsin, pepsin, and α-chymotrypsin	Electron spin resonance technique & Thr-Gly-Gly-Gly-Asn-Val	[36]
Bluefin leatherjacket (*Navodon septentrionalis*) heads	Papain	DPPH radical scavenging activity, hydroxyl radicals and ABTS radicals and Trp-Glu-Gly-Pro-Lys, Gly-Pro-Pro, and Gly-Val-Pro-Leu-Thr	[41]
Bluefin leatherjacket (*Navodon septentrionalis*) skin	Trypsin, flavourzyme, neutrase, papain, alcalase, and pepsin,	DPPH, hydroxyl radicals and oxygen scavenging assays Gly-Ser-Gly-Gly-Leu, Gly-Pro-Gly-Gly-Phe-Ile, and Phe-Ile-Gly-Pro	[99]
Salmon by-product	Alcalase, Flavourzyme, Neutrase, pepsin, Protamex, and trypsin	DPPH and ABTS Phe-Leu-Asn-Glu-Phe-Leu-His-Val	[39]
Horse mackerel (*Magalaspis cordyla*) viscera	*In vitro* gastrointestinal digestion	DPPH and hydroxyl radicals Ala—Cys—Phe—Leu (518.5 Da)	[91]
Giant catfish (Pangasianodon gigas) skin	Visceral alkaline-proteases from Giant catfish, commercial trypsin, Izyme AL^®^	ABTS radical-scavenging, Ferric reducing antioxidant power (FRAP) and metal (ferrous) chelating ability	[100]
Aisan seabass (*Lates calcarifer*) skin	Protease from hepatopancreas of Pacific white shrimp, Alcalase	DPPH and ABTS radical-scavenging activity, Ferric reducing antioxidant power, metal (ferrous) chelating activity, inhibition of lipid peroxidation	[40]
Croaker (Otolithes ruber) skin	Pepsin, trypsin, α-chymotrypsin	Gly-Asn-Arg-Gly-Phe-Ala-Cys-Arg-His-Ala and DPPH and hydroxyl radical-scavenging activity, ferric-reducing antioxidant power, metal (ferrous) chelating activity, inhibition of lipid peroxidation	[91]
Jumbo squid skin gelatin	Enzymatic hydrolysis	Good antioxidant activity for isolated peptide Phe-Asp-Ser-Gly-Pro Ala-Gly-Val-Leu	[101]

**Table 4 marinedrugs-19-00480-t004:** Anticarcinogenic activity of peptides derived from secondary raw materials from fish processing.

By-Product Source for Peptide	Peptide Sequence/Molecular Weight	Anticancer Effect	Researchers
Sepia Ink oligopeptides due to presence of lysine and proline in sequence	N Gln-Pro-Lys with a molecular mass of 343.4 Da	Inhibition of proliferation of human prostate cancer (DU-145) cells	[118]
Tuna dark muscle peptides	Leu-Pro-His-Val-Leu-Thr-Pro-Glu-Ala-Gly-Ala-Thr and Pro-Thr-Ala-Glu-Gly-Gly-Val-Tyr-Met-Val-Thr)	Anticarcinogenic activity against breast cancer cell line	[24]
Snow crab by-product peptides	Two anionic peptides with MW of 537 and 216 Da and three cationic peptides with MW of 228, 241 and 291 Da	anticancer activity on colon, breast, prostate and lung cancer cell lines	[119]
Shrimp shell peptide	Peptides with fractionation size < 10 and 10–30 kDa	Anticancer activity on colon and liver cancer cell lines	[120]
Flathead by-product peptides	<3 kDa	Anticancer activity against HT-29 colon cancer cells up to 91.04%	[121]
*Lates calcarifer* skin peptides	-	Anti-proliferative activity on human colon and liver cancer cell lines	[117]
Flying fish frame peptides	-	Anti-proliferative activity against Hep G2 cells	[122]
Grouper roe peptides	-	Reduced cell viability of oral cancer cells & induced apoptosis of Ca9–22 cells	[123]
Rohu roe peptides	-	Anti-proliferative activity on Human colon cancer cell line	[124]
Threadfin bream (*Nemipterus japonicus*) Back bone peptides	-	Anti-proliferative activity against HepG2 cell lines	[24]
Cuttlefish mantle protein hydrolysates	-	MDA-231 and T47D cancer cell lines with growth inhibition of 78.2 and 66.2%	[125]
Gilthead seabream byproduct peptides	-	Antiproliferative activity on human colon and breast cancer cell lines	[126]

## Data Availability

Not applicable.

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
