# Peer review of "Exploiting of Secondary Raw Materials from Fish Processing Industry as a Source of Bioactive Peptide-Rich Protein Hydrolysates"

_marinedrugs, 2021, doi:10.3390/md19090480_

Round 1

Reviewer 1 Report

abstract line 24: reformulate pls.  However, the value of proteins and other 
nutrients from fish processing by-products became a popular interest due to global demand for

something like : fish processing by products are increasing in value due to global.....

reference no 1 has to be replaced with recent FAO statistics, numbers so old and not relevant . 2018 at least easily available. 

The introduction part on fisheries and discard  has to be updated in general.

Fish feed use should be specified, not included in waste. 

 delete: 216: Antihypertensive drugs like 
captopril and enalapril, sold under the brand names Accupril, Altace, Capoten, Lotensin, Monoril, Prinvil, Vasotec, and Zestril, now regulate blood pressure without addressing the underlying cause, which is yet unknown [31

Author Response

Exploiting of secondary raw materials from fish processing industry as a source of bioactive peptide-rich protein hydrolysates

Manuscript no. – 1323672

Reviewer 1:

  1. Abstract line 24: reformulate pls.  However, the value of proteins and other 
    nutrients from fish processing by-products became a popular interest due to global demand for something like: fish processing by products are increasing in value due to global.....

Response: Whole abstract has been revised and rewritten as;

Developing peptide-based drugs are very promising to address many of the life style mediated diseases which are prevalent in a major portion of the global population. As alternative to synthetic peptide-based drugs, derived peptides from natural sources have gained a greater attention in the last two decades. Aquatic organisms including plants, fish and shellfish are known as a rich reservoir of parent protein molecules which can offer novel sequences of amino acids in peptides having unique bio-functional properties upon hydrolyzing with proteases from different sources. However, rather than exploiting fish and shellfish stocks which are already under pressure due to overexploitation, the processing discards regarded as secondary raw material could be a potential choice for peptide based therapeutic development strategies. In this connection, we have at-tempted to review the scientific reports in this area of research on some of the well-established bioactive properties like antihypertensive, anti-oxidative, anti-coagulative, antibacterial and anticarcinogenic properties with reference to the type of enzymes, substrate used, degree of particular bio-functionality, mechanism, and wherever possible the active amino acid sequences in peptides. Many of the studies have been conducted on hydrolysate (crude mixture of peptides) enriched with low molecular bioactive peptides. In vitro and in vivo experiments on the potency of bioactive peptides to modulate the human physiological functions beneficially have demonstrated that these peptides can be used in the prevention and treatment of non-communicable life style mediated diseases. The information synthesized under this review would serve as a point of reference to drive further the research and development on functionally active therapeutic natural peptides. Availability of such scientific information is expected to open up the new zones of investigations for adding value to the underutilized secondary raw materials which in turn pave way for sustainability in fish processing. However, there are significant challenges ahead in exploring the fish waste as source of bioactive peptides as it demands more studies on mechanisms, structure-functional relationship understanding and clearance from regulatory and statutory bodies before reaching the end user in the form of supplement or therapeutics.

  1. reference no 1 has to be replaced with recent FAO statistics, numbers so old and not relevant . 2018 at least easily available. 

Response: Reference no 1 has been replaced with recent reference. Statistics given in the Manuscript regarding global fish production and its use of food and non-food uses given in SOFIA, 2020 – Publication by FAO is rightly quoted. However, in reference section it was wrongly quoted as FAO, 2012 which is replaced with appropriate reference in the revised MS.

FAO. The State of World Fisheries and Aquaculture (SOFIA)- 2020, Food and Agriculture Organization, Rome, Italy, 2020, 244 p.

  1. The introduction part on fisheries and discard has to be updated in general. Fish feed use should be specified, not included in waste. 

Response: Revised. Use of fish meal in feeds is mentioned in section 1.1. Being a part of review, fisheries and discard are elaborated in Section 1.

  1. delete: 216: Antihypertensive drugs like 
    captopril and enalapril, sold under the brand names Accupril, Altace, Capoten, Lotensin, Monoril, Prinvil, Vasotec, and Zestril, now regulate blood pressure without addressing the underlying cause, which is yet unknown [31

Response: Deleted

Reviewer 2 Report

The manuscript “Exploiting of secondary raw materials from fish processing industry as a source of bioactive peptide-rich protein hydrolysates” should be improved because there are several sentences not clear. It is also recommended that it be shortened due to many repeated sentences.

Abstract

Lines 19-36 – This abstract is very generic and does not provide any details about the content of this manuscript.

Figure 1 – Are crustaceans’ viscera amounts very significant? Which type of bones, squid pens or cuttlefish bones?

Lines 52-54 – Please revise the sentence “Wider attention… environmental” for clarification.

Line 55 – I suggest “biochemicals”.

Line 70 – Is it protein derived? Please check.

Line 73 – I suggest replacing “originating” by “originated”.

Lines 73 and 74 – The article of reference [11] doesn’t deal with the anticoagulant and antiplatelet effects of bioactive peptides. Please check.

Line 80 – Please clarify “functioning system”.

Line 82 – I suggest replacing “produced” by another more adequate word.

Line 83 – I also suggest revising “starting… polypeptides” for improvement.

Lines 87-89 – This was previously mentioned in lines 70 and 71.

Lines 92-94 – Please revise the sentence “By adding water… protein hydrolysates” because it doesn’t seem correct.

Line 97 – I suggest replacing “created” by “produced”, for instance.

Line 98 and 99 – Please revise this sentence because it is not clear.

Lines 102 and 103 – This sentence seems irrelevant and in some way is contradictory with the sentence in lines 98 and 99.

Lines 104-112 – These sentences mention the types of proteases used but this is not the hydrolysis mechanism.

Line 114 – I suppose that it is not “recover” but “release”.

Line 117 – Is it “peptide sequence” or “amino acid sequence in peptides” Please check.

Line 126 – Why this particular quotation? There are many works where several enzymes were used.

Line 128 – The amino acid composition of a peptide and the sequence of these amino acids determine all biological activities of peptides, not only the antioxidant capacity, I suppose.

Line 129 – I think the degree of hydrolysis is important to evaluate the hydrolysis process but what determines the biological activities of peptides are their amino acid composition, sequence of amino acids and size/molecular weight.

Lines 134 and 135 – I think that in protein hydrolysates there are no proteins.

Line 136 – I suggest replacing “molecular” by “peptidic”.

Line 137 – Please clarify what is meant by “high-end by-products”.

Lines 138 and 139 – This sentence is a repetition of others.

Line 147 and 148– Please clarify this sentence.

Line 166-169 – This sentence seems irrelevant and a repetition of previous sentences.

Line 187-190 – These are more repetitions.

Line 210-212 – Please clarify this information.

Line 213 – I suggest replacing “changes” by “converts”.

Line 214 – The amino acid sequence of Ang I is not necessary to be repeated.

Line 230 – I suppose that it is “showing” and not “showed”.

Lines 245 and 246 – Please consider this alternative: “Peptides exhibiting ACE-I inhibitory activity were specially dipeptides such as: glycine,… threonine or valine with proline; glycine-isoleucine; and phenylalanine-asparagine”.

Line 258-275 – This introduction is too long and the mechanisms of oxidation have been described in multiple works.

Line 277-304 – This second introduction is also too long.

Line 327 – I suppose that it is “atomic hydrogen” and not “proton”.

Lines 345-347 – The subject of this article was to evaluate the natural antibacterial and haemolytic activities in three fish species.

Line 360 – Please revise “water circumstances”.

Lines 370-379 – I suggest the scientific names in italics.

Lines 375 and 376 – Please revise the sentence “suggesting… action” for improvement.

Lines 389 and 390 – It is “amino acids” and not “peptides” and “molecular weight below 10 kDa”. Please check.

Lines 396 and 397 – The sentence “chemical or biochemical… (FPH)” seems irrelevant.

Line 442 – Is it “clarification”?

Line 443 – Please improve the sentence “aquatic sources may be…”

Line 482 – It is “gelatin”.

Lines 504-506 – Please revise the sentence “When it comes… to be established” for clarification.

Line 380 – Please revise this reference.

Author Response

Exploiting of secondary raw materials from fish processing industry as a source of bioactive peptide-rich protein hydrolysates

Manuscript no. – 1323672

Reviewer 2:

  1. Abstract Lines 19-36 – This abstract is very generic and does not provide any details about the content of this manuscript.

Response: Whole abstract is rewritten wherein all suggestions by the reviewer have been taken care of. 

  1. Figure 1 – Are crustaceans’ viscera amounts very significant? Which type of bones, squid pens or cuttlefish bones?

Response: In the Figure 1, waste generation is categorized in general irrespective of their significant portions. The term ‘bones’ is used in general here to represent cuttlebones or squid pens.

  1. Lines 52-54 – Please revise the sentence “Wider attention… environmental” for clarification.

Response: The sentence has been revised as;

Utilization of these material could improve the economics and sustainable at the same time.

  1. Line 55 – I suggest “biochemicals”.

Response: revised

  1. Line 70 – Is it protein derived? Please check.

Response: proteins, derived will be correct

  1. Line 73 – I suggest replacing “originating” by “originated”.

Response: revised

  1. Lines 73 and 74 – The article of reference [11] doesn’t deal with the anticoagulant and antiplatelet effects of bioactive peptides. Please check.

Response: The reference was wrongly quoted here. Correct reference is quoted in the revision. Reference No [67].

  1. Line 80 – Please clarify “functioning system”.

Response: Functioning system include various cardiovascular, nervous, gastrointestinal and immune system for body functioning.

The line has been revised as; system including cardiovascular, nervous, gastrointestinal and immune system

  1. Line 82 – I suggest replacing “produced” by another more adequate word.

Response: Formed would be more adequate word and the word produced is replaced with a word formed

  1. Line 83 – I also suggest revising “starting… polypeptides” for improvement.

Response: Revised as suggested by reviewer as; including peptides having two or more amino acids.

  1. Lines 87-89 – This was previously mentioned in lines 70 and 71.

Response: We agree with the reviewer. Line 87-89 is deleted in the revised version.

  1. Lines 92-94 – Please revise the sentence “By adding water… protein hydrolysates” because it doesn’t seem correct.

Response: The sentence has been revised as;

Hydrolytic breakdown of high molecular weight proteins to low molecular weight proteins is the basis in protein hydrolysate production.

  1. Line 97 – I suggest replacing “created” by “produced”, for instance.

Response: revised

  1. Line 98 and 99 – Please revise this sentence because it is not clear.

Response: The sentence is deleted for better clarity in the review paper.

  1. Lines 102 and 103 – This sentence seems irrelevant and in some way is contradictory with the sentence in lines 98 and 99.

Response: Lines 98 and 99 are deleted

  1. Lines 104-112 – These sentences mention the types of proteases used but this is not the hydrolysis mechanism.

Response: Whole paragraph is revised and mechanism of hydrolysis is added in the revision. Few sentences have been modified in the revision.

  1. Line 114 – I suppose that it is not “recover” but “release”.

Response: revised

  1. Line 117 – Is it “peptide sequence” or “amino acid sequence in peptides” Please check.

Response: Rewritten as amino acid sequence in peptides

  1. Line 126 – Why this particular quotation? There are many works where several enzymes were used.

Response: The statement “Many bioactive peptides have been created in the lab utilising a variety of commercial proteases” has been deleted. It is rewritten as “There are many scienitifc reports where different proteases from various sources have been used to produce bioactive peptides with desired health beneficial properties”.

  1. Line 128 – The amino acid composition of a peptide and the sequence of these amino acids determine all biological activities of peptides, not only the antioxidant capacity, I suppose.

Response: Agreed with the reviewer suggestion. The sentence has been rewritten in the revised manuscript.

  1. Line 129 – I think the degree of hydrolysis is important to evaluate the hydrolysis process but what determines the biological activities of peptides are their amino acid composition, sequence of amino acids and size/molecular weight.

Response: We are sorry to differ with the reviewer that the degree of hydrolysis is not only important to evaluate the process. But also, for the given enzyme, substrate and the hydrolysis conditions, the composition, sequence and size/molecular weight are also dictated by the parameter called degree of hydrolysis.  The same statements have been added in the revised manuscript.

  1. Lines 134 and 135 – I think that in protein hydrolysates there are no proteins.

Response: We agree with the reviewer and rewritten the statement in the revised manuscript.

  1. Line 136 – I suggest replacing “molecular” by “peptidic”.

Response: revised

  1. Line 137 – Please clarify what is meant by “high-end by-products”.

Response: We used the phrase ‘high-end’ to convey the meaning of high value of the product and for functional food uses. However, we understood that we failed in conveying the message. We have rewritten the statement as” Hydrolysis of proteins converts the secondary raw materials into a high value product which is rich in amino acids and biologically active peptides.

  1. Lines 138 and 139 – This sentence is a repetition of others.

Response: We are sorry to express that we couldn’t understand what the reviewer means. In the present manuscript no such statement is repeated.

  1. Line 147 and 148– Please clarify this sentence.

Response: The sentences have been rewritten for better clarity.

  1. Line 166-169 – This sentence seems irrelevant and a repetition of previous sentences.

Response: As suggested by the reviewer, the sentences (line number 166-169) have been deleted in the revised manuscript.

  1. Line 187-190 – These are more repetitions.

Response: The repetitions have been removed in the revised manuscript

  1. Line 210-212 – Please clarify this information.

Response: Line number 210-212 have been removed for better clarity

  1. Line 213 – I suggest replacing “changes” by “converts”.

Response: revised

  1. Line 214 – The amino acid sequence of Ang I is not necessary to be repeated.

Response: revised

  1. Line 230 – I suppose that it is “showing” and not “showed”.

Response: revised

  1. Lines 245 and 246 – Please consider this alternative: “Peptides exhibiting ACE-I inhibitory activity were specially dipeptides such as: glycine,… threonine or valine with proline; glycine-isoleucine; and phenylalanine-asparagine”.

Response: revised

  1. Line 258-275 – This introduction is too long and the mechanisms of oxidation have been described in multiple works.

Response: Though it has been explained in others work, for the readability and clarity of the manuscript, we would like to retain the description on mechanism of oxidation as this section describes anti-oxidative properties of fish waste protein hydrolysate.

  1. Line 277-304 – This second introduction is also too long.

Response: We have shortened the introduction in the revised manuscript

  1. Line 327 – I suppose that it is “atomic hydrogen” and not “proton”.

Response: revised                  

  1. Lines 345-347 – The subject of this article was to evaluate the natural antibacterial and haemolytic activities in three fish species.

Response: The sentences from 345-347 have been removed in the revised manuscript as they are not relevant as correctly pointed out by the reviewer

  1. Line 360 – Please revise “water circumstances”.

Response: revised as ‘water’

  1. Lines 370-379 – I suggest the scientific names in italics.

Response: revised

  1. Lines 375 and 376 – Please revise the sentence “suggesting… action” for improvement.

Response: revised as- implicating the action of amino acid sequence

  1. Lines 389 and 390 – It is “amino acids” and not “peptides” and “molecular weight below 10 kDa”. Please check.

Response: Corrected in the revised manuscript as suggested by the reviewer

  1. Lines 396 and 397 – The sentence “chemical or biochemical… (FPH)” seems irrelevant.

Response: The sentences have been removed in the revised manuscript

  1. Line 442 – Is it “clarification”?

Response: revised as ‘calcification

  1. Line 443 – Please improve the sentence “aquatic sources may be…”

Response: Aquatic source is one of the best alternatives to milk

  1. Line 482 – It is “gelatin”.

Response: revised

  1. Lines 504-506 – Please revise the sentence “When it comes… to be established” for clarification.

Response: The sentence has been revised for better clarity.

  1. Line 380 – Please revise this reference.

Response: It is correctly quoted for snow crab antibacterial activity.

Round 2

Reviewer 1 Report

line 63 reformulate :Currently, a large part of the total fish production is discarded or processed to fishmeal

line 108-114 missing reference

line 123-139 Needed text ? all based on casein ref 40 years old !

line 145 are all factors to consider reformulate: are important.

line 146 reformulate :have to be optimized

line 154  correct: used chymotrypsin, papain, neutrase, and trypsin at the appropriate pH and temperature for each enzyme

chymotrypsin, papain, neutrase, and trypsin at specific pH and temperature for each enzyme is used

line 171 bioactive compounds in food/feed and....

line 172  delete: bioactive components in industrial and (not relevant)

line 177 replace creatures with organisms ; delete good

line 199 new class of bioactive : replace with : a novel source of ....

line 204  replace thrown with discarded 

line 217  delete: It is not possible to include all the studies as it will be very exhaustive. 

line 265 replace were with are

line 271  replace had with showed

the 4.2.1. and 4.2.2. chapter can be deleted

line 360 delete:  which work ,  replace with comma ..., modifying....

line 375 delete : Scomber scombrus , latin name only fisrst time !

line 415 apoptosis, antiproliferative, stimulated, , have to be in the same grammatic form :::   for ex : apoptosis, antiproliferation, stimulation 

line 415- please reformulate, no good sentence !!!! Free amino acids and peptides generated were found to be low molecular weight (300-1950 Da) imparting them higher mobility and diffusivity may be responsible for their anticancer activity [130,131]

line 470 delete; from collagen

line 476 delete: fish

line 479 delete: higher

line 483 delete : better

line 485  delete good

line 490 delete better

line 494 reformulate, suggestion follows : The current interest for fish hydrolysates containing large amount of protein and thereby peptides and amino acids in the  is of large interest and importance for future research. 

Author Response

Reply to the Review Report (Reviewer 1)

Manuscript ID: marinedrugs-1323672

We would like to thank the Reviewer for his/her constructive points to improve the scientific value of the paper. We appreciate the time and effort he/she has sacrificed to perform the review. We have corrected the paper according to reviewer’s comments. Red colour in the text indicates the corrections or modifications in the manuscript.

Bellow, we are presenting detailed responses to each comment of the Reviewer.

Reviewer 1:

  1. line 63 reformulate: Currently, a large part of the total fish production is discarded or processed to fishmeal

Response: suggested changes have been made

  1. line 108-114 missing reference

Response: The sentence has been supported by the work of

Naqvi, N.; Liu, K.; Graham, R.M.; Husain, A. Molecular basis of exopeptidase activity in the C-terminal domain of human angiotensin I-converting enzyme: insights into the origins of its exopeptidase activity. J Biol Chem. 2005; 280(8):6669-6675.

  1. line 123-139 Needed text ? all based on casein ref 40 years old !

Response: We agree that lines 123-139 mentioned here are older. Though it has been explained in others work, for the readability and clarity of the manuscript, we would like to retain the description on mechanism of hydrolysis as suggested by another reviewer.

  1. line 145 are all factors to consider reformulate: are important.

Response: The modifications have been made according to suggestion.

  1. line 146 reformulate :have to be optimized

Response: The changes have been modified as suggested.

  1. line 154  correct: used chymotrypsin, papain, neutrase, and trypsin at the appropriate pH and temperature for each enzyme chymotrypsin, papain, neutrase, and trypsin at specific pH and temperature for each enzyme is used

Response: The modifications have been made according to suggestion.

  1. line 171 bioactive compounds in food/feed and....

Response: Revised as suggested

  1. line 172 delete: bioactive components in industrial and (not relevant)

Response: Corrected as suggested

  1. line 177 replace creatures with organisms; delete good

Response: Revised as suggested

  1. line 199 new class of bioactive : replace with : a novel source of ....

Response: Modified as suggested

  1. line 204  replace thrown with discarded 

Response: Revised as suggested

  1. line 217  delete: It is not possible to include all the studies as it will be very exhaustive. 

Response: Corrected as suggested

  1. line 265 replace were with are

Response: Revised as suggested

  1. line 271  replace had with showed

Response: Modified as suggested

  1. the 4.2.1. and 4.2.2. chapter can be deleted

Response: Sections 4.2.1 and 4.2.2 have been shortened as per the suggestion of the reviewer in the first revision. Though it has been explained in others work, for the readability and clarity of the manuscript, we would like to retain the description on antioxidants, mechanism of oxidation as this section describes anti-oxidative properties of fish waste protein hydrolysate.

  1. line 360 delete:  which work ,  replace with comma ..., modifying....

Response: Corrected as suggested

  1. line 375 delete : Scomber scombrus , latin name only fisrst time !

Response: Revised as suggested

  1. line 415 apoptosis, antiproliferative, stimulated, , have to be in the same grammatic form :::   for ex : apoptosis, antiproliferation, stimulation 

Response: Modified as suggested

  1. line 415- please reformulate, no good sentence !!!! Free amino acids and peptides generated were found to be low molecular weight (300-1950 Da) imparting them higher mobility and diffusivity may be responsible for their anticancer activity [130,131]

Response: The statement has been revised as,

The lower molecular weight of free amino acids and peptides (300-1950 Da) imparted them anti-cancerous activity, due to high mobility and diffusion

  1. line 470 delete; from collagen

Response: Corrected as suggested

  1. line 476 delete: fish

Response: Revised as suggested

  1. line 479 delete: higher

Response: Modified as suggested

  1. line 483 delete: better

Response: Revised as suggested

  1. line 485 delete good

Response: Changed as suggested

  1. line 490 delete better

Response: Revised as suggested

  1. line 494 reformulate, suggestion follows: The current interest for fish hydrolysates containing large amount of protein and thereby peptides and amino acids in the is of large interest and importance for future research. 

Response: Statement has been reformulated as mentioned below,

The current interest for fish hydrolysates containing large amount of protein and thereby peptides and amino acids in the researchers is of large interest and importance for future research.

Reviewer 2 Report

The second version of the manuscript “Exploiting of secondary raw materials from fish processing industry as a source of bioactive peptide-rich protein hydrolysates” was clarified and considerably improved. However, some minor suggestions of changes are included below.

Abstract

Line 28 – It is “attempted”.

Introduction

Line 59 – Please consider this alternative: “…on the utilization of these materials could improve the economics and sustainability at the…”

Line 75 – I suppose that it is “…in hydrolysates derived…”

Lines 153 and 154 – Please check quotation 23 because this paper is on peptides from bullfrog muscle proteins.

Line 161 – Please revise this sentence because it is not clear.

Bioactivities of fish waste protein hydrolysates

Table 2. The quotation 61 in line on “Tuna heads” is on Sardinella hydrolysates. Please check.

Table 2. The fish species studied in quotation 81 is not Capros aper, but Mustelus mustelus. Please check.

Table 3. Why the repetition of quotation 97?

Table 3. The hydrolysates from croaker (Otolithes ruber) skin were studied in the paper with quotation 90. They were studied by the same authors but published in a paper in 2012. Please check.

Table 3. I suppose that the last quotation in this table is 101. Please check.

Lines 376 and 380 – Innocua is in italics.

Line 385 – There is a repetition of quotation 100. Please check.

Line 389 – I suggest “… is found in peptides which are released…” Please check.

Line 390 – It is “…50 amino acids…”

Table 4. I think that quotation 124 in the line “Rohu roe peptides” is 78. Please check.

Table 4. Please check the number of reference 118 on “threadfin bream peptides”.

Lines 452-454 – The subject of quotation 145 doesn’t fit in the scope of the current work.

Line 504 – Please revise the sentence “However, the food safety… details” for clarification.

References

The specific epithet of species scientific names should be in lowercase letters.

Why the authors’ names in references 117 and 157 are in capital letters?

The reference 82 seems incomplete. Please check.

Author Response

Reply to the Review Report (Reviewer 2)

Manuscript ID: marinedrugs-1323672

We would like to thank the Reviewer for his/her constructive points to improve the scientific value of the paper. We appreciate the time and effort he/she has sacrificed to perform the review. We have corrected the paper according to reviewer’s comments. Red colour in the text indicates the corrections or modifications in the manuscript.

Bellow, we are presenting detailed responses to each comment of the Reviewer.

Reviewer 2:

The second version of the manuscript “Exploiting of secondary raw materials from fish processes industry as a source of bioactive peptide-rich protein hydrolysates” was clarified and considerably improved. However, some minor suggestions of changes are included below.

Response: We are thankful to the reviewer for the positive and constructive comments.

  1. Abstract Line 28 – It is “attempted”.

Response: Modified as suggested

  1. Introduction Line 59 – Please consider this alternative: “…on the utilization of these materials could improve the economics and sustainability at the…”

Response: Modified as suggested

  1. Line 75 – I suppose that it is “…in hydrolysates derived…”

Response: The suggested corrections have been made

  1. Lines 153 and 154 – Please check quotation 23 because this paper is on peptides from bullfrog muscle proteins.

Response: Thank you for the careful evaluation. The details have been corrected

  1. Line 161 – Please revise this sentence because it is not clear. Bioactivities of fish waste protein hydrolysates

Response: Statement has been revised as,

However, for the given enzyme, substrate and the hydrolysis conditions- the composition, sequence and size/molecular weight of derived peptides in turn the bio-functionalities of protein hydrolysates derived from fish processing waste are all dictated by DH.

  1. Table 2. The quotation 61 in line on “Tuna heads” is on Sardinella Please check.

Response: The reported reference is correct and describes as yellow sole frame

  1. Table 2. The fish species studied in quotation 81 is not Capros aper, but Mustelus mustelus. Please check.

Response: The reported reference is correct and describes Hound viscera

  1. Table 3. Why the repetition of quotation 97?

Response: Thank you for the careful evaluation. The repetition has been removed

  1. Table 3. The hydrolysates from croaker (Otolithes ruber) skin were studied in the paper with quotation 90. They were studied by the same authors but published in a paper in 2012. Please check.

Response: Thank you for the careful evaluation. The reference has been corrected.

  1. Table 3. I suppose that the last quotation in this table is 101. Please check.

Response: Thank you for the careful evaluation. The details have been corrected

  1. Lines 376 and 380 – Innocua is in italics.

Response: Modified as suggested

  1. Line 385 – There is a repetition of quotation 100. Please check.

Response: Details has been modified

  1. Line 389 – I suggest “… is found in peptides which are released…” Please check.

Response: Modified as suggested

  1. Line 390 – It is “…50 amino acids…”

Response: Modified as suggested

  1. Table 4. I think that quotation 124 in the line “Rohu roe peptides” is 78. Please check.

Response: The details have been modified

  1. Table 4. Please check the number of reference 118 on “threadfin bream peptides”.

Response: Thank you for the careful evaluation. The details have been corrected.

  1. Lines 452-454 – The subject of quotation 145 doesn’t fit in the scope of the current work.

Response: The details have been updated

  1. Line 504 – Please revise the sentence “However, the food safety… details” for clarification.

Response: The statement has been revised as,

However, there are limited studies on safety aspects of hydrolysates produced from fish processing waste and further research is required in the same line.

  1. References The specific epithet of species scientific names should be in lowercase letters.

Response: The details have been modified through the references section

  1. Why the authors’ names in references 117 and 157 are in capital letters?

Response: The details have been modified

  1. The reference 82 seems incomplete. Please check.

Response: The reference has been completed.